# FASTER REINFORCEMENT LEARNING WITH VALUE TARGET LOWER BOUNDING

## ABSTRACT

We show that an arbitrary lower bound of the optimal value function can be used to improve the Bellman value target during value learning. In the tabular case, value learning using the lower bounded Bellman operator converges to the same optimal value as using the original Bellman operator, at a potentially faster speed. In practice, discounted episodic return from the training experience or discounted goal return from hindsight relabeling can serve as the value lower bound when the environment is deterministic. We experiment on Atari games, FetchEnv tasks and a challenging physically simulated car push and reach task. We show that in most cases, simply lower bounding with the discounted episodic return performs better or as well as common baselines such as TD3, SAC and Hindsight Experience Replay (HER). It learns much faster than TD3 or HER on some of the harder continuous control tasks, requiring minimal additional computation and no parameter tuning. We are not the first to introduce this simple yet effective technique, but the first to demonstrate its optimality in theory and effectiveness in a wide range of tasks and related baseline methods.

## 1 INTRODUCTION

In temporal difference (TD) learning, the value function is adjusted toward its Bellman target, which is the reward of the current step plus the discounted value of the next state. This forms the basis of many state of the art reinforcement learning (RL) algorithms such as DQN (Mnih et al., 2013), DDPG (Lillicrap et al., 2015), TD3 (Fujimoto et al., 2018), and SAC (Haarnoja et al., 2018).

The value of the next state is typically estimated using a "bootstrapped value" based on the value function itself, which is being actively learned during training. The bootstrapped values can be random or very inaccurate, especially at the initial stage of training. Consequently, the Bellman value targets as well as the learned value are usually far away from the optimal value.

Naturally, this leads to the following idea: If we can make the value target closer to the optimal value, we may speedup TD learning. For example, we know that the optimal value is just the expected discounted return of the optimal policy, which always upper bounds the expected return of any policy. For episodic RL tasks, we could use the observed discounted return up to episode end from the training trajectories to lower bound the value target. This makes the new value target closer to the optimal value, when the empirical return is higher than the Bellman target.

Will such a way of lower bounding the value target work: Will it still converge? Will it converge to the optimal value? Will it speed up value learning?

## 2 THEORETICAL RESULTS FOR THE TABULAR CASE

For the tabular case, value target lower bounding converges to the same optimal value as the original Bellman value learning, and the proof is also straightforward.

### 2.1 BACKGROUND

In finite MDPs with a limited number of states and actions, a table can be used to keep track of the value of each state. Using dynamic programming algorithms such as value iteration, values

are guaranteed to converge to the optimal through Bellman updates (Chapter 4.4 (Sutton & Barto, 2018)).

---

**Algorithm 1:** Bellman value iteration with value target lower bounding

**Data:** Finite MDP $p(s', r|s, g, a)$, convergence threshold $\theta$

**Result:** State value $v(s)$

1 $v(s) \leftarrow 0$;
2 **repeat**
3     $\Delta \leftarrow 0$;
4     **for** *each state s* **do**
5         $v \leftarrow v(s)$;
6         $v(s) \leftarrow \max(f, \max_a \sum_{s', r} p(s', r|s, g, a)[r + \gamma v(s')])$;
7         $\Delta \leftarrow \max(\Delta, |v(s) - v|)$;
8     **end**
9 **until** $\Delta < \theta$;

---

The core of the algorithm is the Bellman update of the value function, $\mathcal{B}(v)$:

$$\mathcal{B}(v)(s) \coloneqq \max_a \sum_{s', r} p(s', r|s, g, a)[r + \gamma v(s')] \tag{1}$$

It is well known that the Bellman operator, $\mathcal{B}$, is a contraction mapping over value functions (Denardo, 1967). That is, for any two value functions $v_1$ and $v_2$, $|\mathcal{B}(v_1) - \mathcal{B}(v_2)| \leq \gamma|v_1 - v_2|$ for the discount factor $\gamma \in [0, 1)$. This guarantees that any value function under the algorithm converges to the optimal value.[1]

## 2.2 VALUE TARGET LOWER BOUNDING CONVERGENCE THEOREM

**Theorem 1.** *Suppose the optimal value under the Bellman operator is $\mathcal{B}^\infty(v)$. For any value function $f$ that lower bounds the optimal value, i.e. $\forall s, f(s) \leq \mathcal{B}^\infty(v)(s)$, if we define the lower bounded Bellman operator as $\mathcal{M}_f \circ \mathcal{B}(v) \coloneqq \max(\mathcal{B}(v), f)$, then $(\mathcal{M}_f \circ \mathcal{B})^\infty(v)$ converges to $\mathcal{B}^\infty(v)$.*

A few things to note about the proof (see Appendix A.1).

First, this only proves convergence, not contraction under the original $||v_1 - v_2||_\infty$ metric. In the case of the Bellman operator, contraction shows that $\forall v_1, v_2$ value functions, $||\mathcal{B}(v_1) - \mathcal{B}(v_2)||_\infty \leq \gamma||v_1 - v_2||_\infty$. Here, for value target lower bounding, there can be counter examples where $\mathcal{M}_f \circ \mathcal{B}$ does not always contract in the original metric space for value functions. Here, convergence relies on the convergence of the Bellman value iteration and the existence of the fixed point $v^*$. One difficulty caused by this change is that the stopping criterion in Algorithm 1 ($\Delta < \theta$) no longer works, as we do not have access to the converged value during learning. This is perhaps not a serious concern in practice, as people often train algorithms for a fixed number of iterations or time steps.

Second, based on the proof, the new algorithm is at least as fast as the original. When the lower bound actually improves the value target, i.e. $f(s) > \mathcal{B}(v_1)(s)$, there is a chance for the convergence to be faster. Convergence is strictly faster when the lower bound $f$ has an impact on the $L_\infty$ distance between the current value and the optimal value, i.e. it increases the value target for the states where the differences between the value target and the optimal value are the largest.

Third, the lower bound function doesn't have to be static during training. As long as there is a single $f$ during each iteration, convergence property is preserved.

Fourth, the theory works even when the underlying MDP is stochastic. Only the lower bounds based on empirical return introduced below require the MDP to be deterministic.

---

[1]See, for example, page 8 for the gist of the proof `https://people.eecs.berkeley.edu/~pabbeel/cs287-fa09/lecture-notes/lecture5-2pp.pdf`

## 3 EXAMPLE LOWER BOUND FUNCTIONS

We show a few cases where lower bound functions can be readily obtained from the training experience. Future work may investigate alternative lower bounds.

### 3.1 EPISODIC TASKS

In episodic tasks, discounted return is only accumulated up to the last step of an episode. In this case, we can wait until an episode ends, and compute future discounted returns of all time steps inside the episode. This discounted return is guaranteed to be a lower bound of the optimal value, if the environment is deterministic, i.e. the reward sequence can be repeated using the exact same sequence of actions. (The behavior policy need not be deterministic, as long as the policy class contains the deterministic optimal policy.) To make training efficient, we can compute and store such discounted returns into the replay buffer for each time step, and simply read them out during training.

We call this variant lb-DR, short for lower bounding with discounted return.

#### 3.1.1 EPISODIC WITH HINDSIGHT RELABELED GOALS

In goal conditioned tasks, one helpful technique is hindsight goal relabeling (Andrychowicz et al., 2017). It takes a future state that is $d$ time steps away from the current state as the hindsight / relabeled goal for the current state. When the goal is reached, a reward of 0 is given, otherwise a -1 reward is given for each time step.

In this case, we know it took $d$ steps to reach the hindsight goal, so the discounted future return is:

$$R_d = \sum_{i=0,..,d-1} -1\gamma^i$$
$$= -1(1-\gamma^d)/(1-\gamma) \tag{2}$$

This calculation can be done on the fly as hindsight relabeling happens, requiring no extra space and very little computation.

We call this variant lb-GD, short for lower bounding with goal distance based return.

Additionally, we can also apply lb-DR and lb-GD together, with discounted return lower bounding (lb-DR) on the original experience and goal distance return lower bounding (lb-GD) on the hindsight experience, giving the lb-DR+GD variant, which was used by Fujita et al. (2020) independently.

### 3.2 NON-EPISODIC TASKS WITH POSITIVE REWARDS

When the task is continuing, without an episode end, discounted return needs to be accumulated all the way to infinity. This makes it difficult to lower bound the value if rewards can be negative. When rewards are always non-negative, one can still use the discounted return of the future n-steps to lower bound the value. Chapter 3.3 of Sutton & Barto (2018) has more details on episodic vs continuing tasks.

## 4 INTEGRATION INTO RL ALGORITHMS

### 4.1 BACKGROUND

The value target lower bounds can be readily plugged into RL algorithms that regresses value to a target, e.g. DQN, DDPG or SAC.

In these algorithms, the action value $q(s, a)$ is learned through a squared loss with the target value $y$. In one step TD return, for a batch $\mathbf{B}$ of experience $\{s, a \to r, s'\}$, the loss is:

$$\mathcal{L}_q := \sum_{(s,a,r,s') \in \mathbf{B}} |q(s, a) - y|^2 \tag{3}$$

In one step TD return, $y$ is the one step TD return $\hat{q}(s, a, r, s')$:

$$\hat{q}(s, a, r, s') := r(s, a) + \gamma q'(s', \mu'(s')) \tag{4}$$

Here, $q'$ and $\mu'$ are the bootstrap value and policy functions, typically following the value and policy functions in a delayed schedule during training. (They are also called "target value" and "target policy", and are very different from the "value target" $y$ in this paper.)

## 4.2 VALUE TARGET LOWER BOUNDING

With lower bounding, we replace the value target $y$ with the lower bounded target:

$$y \leftarrow \max(f, \hat{q}(s, a, r, s')) = \max(f, r + \gamma q'(s', \mu'(s'))) \tag{5}$$

This is subtly but importantly different from lower bounding the $q$ value directly (Oh et al., 2018; Tang, 2020): $q(s, a) \leftarrow \max(f, q(s, a))$, which stays overestimated if $q(s, a)$ initially overestimates. This is the same as was done by Fujita et al. (2020) (confirmed via personal communication).

This way of simply lower bounding the value target does not require any tuning parameter, but one can always interpolate between these two value targets using a mixing weight $\alpha$:

$$y \leftarrow (1 - \alpha)\hat{q}(s, a) + \alpha \max(f, \hat{q}(s, a)) \tag{6}$$

A small $\alpha$ dampens the effect of the new value target, and may be desirable in practice when assumptions of the theorem can be violated, e.g. for non-deterministic tasks.

See Appendix A.2 for an illustrative example of how value target lower bounding works in practice.

## 5 EXPERIMENTS

The goal is to demonstrate the sample efficiency of lower bounding the value target over baseline such as DDPG, TD3, SAC and HER. Because the lower bounded value target can now look potentially many steps into the future, we suspect it to be best suited for long horizon, sparse reward tasks. Hence, we choose to experiment on the following tasks.

### 5.1 ENVIRONMENTS AND TASKS

We experiment on three sets of tasks with different input characteristics and control difficulty. Some of the tasks are not goal conditioned, so only lower bounding with empirical discounted return is available. Some of them are goal conditioned, so both empirical discounted return and hindsight relabeling with discounted goal return as lower bound are available.

#### 5.1.1 ATARI GAMES

We experiment on the classical Atari games with image input to test using discounted episodic return to lower bound value target. We picked the popular games Breakout, Seaquest, Space Invaders, Atlantis, Frostbite and Q*bert, and only experimented on them. As with prior work (Oh et al., 2018), we evaluate on the deterministic versions of the games, NoFrameskip-v4 with actions repeated for a fixed (four) frames.

#### 5.1.2 EPISODIC FETCH PUSH, SLIDE AND PICKANDPLACE

The FetchEnv tasks (Plappert et al., 2018) are goal conditioned tasks with a robotic arm moving objects on a table. Robot states and object position serve as input. The agent outputs continuous actions taking the form of relative positions to move to. A PID controller translates the relative position actions into the exact torque applied at each joint. Rewards are sparse and goal-conditioned, with -1 for non-goal states and 0 for goal states.

By default the FetchEnv tasks are non-episodic. They reset every 50 steps, but all steps including the step right before task reset have the same positive discount (Andrychowicz et al., 2017). As explained in Section 3.1, to allow reliable estimates of return lower bounds to be calculated from past experience, we make them episodic by adding a gym wrapper around the environment to end an episode after its goal is achieved, and reset the task. When a goal is not reached within 50 steps, we just reset the task without ending the episode, as is done in the original FetchEnv, and such experience is not used in value target lower bounding.[2] This also changes the nature of the tasks, so the agent does not have to stay at the goal state indefinitely, but instead only needs to reach the goal position as fast as possible. This makes the episodic FetchEnv tasks slightly easier to train than the original tasks, because the agent only needs to reach the goal state quickly, instead of having to reach and stay at the goal position indefinitely. (There are ways to avoid changing the desired behavior by e.g. including agent's speed into the goal state or requiring the agent to stay at the goal position for several time steps before ending the episode. This seems orthogonal to the main idea here, and is not included in this work.)

Compared with the Atari games, the inputs are simpler, no longer image based, but the control task is continuous, under realistic physical simulation and harder.

### 5.1.3 PIONEER PUSH AND REACH TASKS

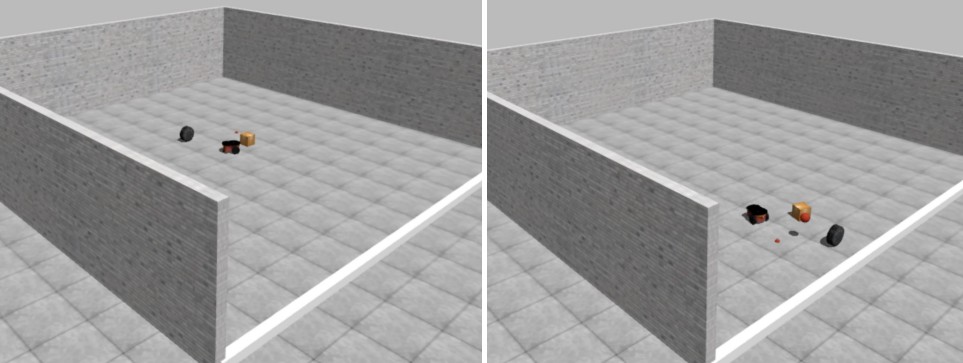

Figure 1: The Pioneer Push task and the Push and Reach task.

This is a set of challenging goal reaching and object pushing tasks for the physically simulated car Pioneer 2dx. The car is 0.4 meter long. Objects and goal positions are randomly initialized between 0.5 meter to 1 meter of each other inside a 10 meter by 10 meter flat space. Inputs are the car and object states and the goal positions, and actions are the forces applied on the two driving wheels.

For the Pioneer Push task, the car has to push a block to within 0.5 meter of the 2 dimensional goal position indicated by a small red dot on the ground. For the Pioneer Push and Reach task, the car has to first push the object to the goal location (red dot) and then drive to a separate goal position (red ball in the air); the goal is achieved when the concatenation of the two goal locations (for Push and for Reach) is within 0.5 of the concatenated achieved positions (of the block and the car) in $L_2$-distance.

Similar to FetchEnv, we make the tasks episodic with sparse goal reward.

These tasks take longer time to accomplish, and also take longer time to train than the FetchEnv tasks. Some of the reasons are the force based wheel control instead of the higher level position control, and the arena space being much larger than just a tabletop.

---

[2] Fujita et al. (2020) chose to end episodes when either a maximum of T time steps is reached or the goal is reached, and provided the agent with the number of timesteps since episode start as input to the agent, so that the agent is aware of the approaching episode end.

## 5.2 BASELINES

Baselines include DDPG (Lillicrap et al., 2015), TD3 (Fujimoto et al., 2018), SAC (Haarnoja et al., 2018) and HER (Andrychowicz et al., 2017). Implementations are based on open sourced repositories, and baseline performance is verified against published results under similar settings.

## 5.3 HYPERPARAMETERS

The value target lower bounding method itself does not have any hyperparameter, the only hyperparameters come from the baseline method. Hyperparameters for the baselines follow published work as much as possible. When tuning baseline hyperparameters, we searched for the best performance in totoal episodic reward, on one set of random seeds. Optimal hyperparameters are then fixed and evaluated on a separate set of random seeds never seen during development. For the treatment, we just used the optimal parameters from the baseline tuning, except for the Atari games where we found the treatment to benefit from more (eight) minibatch updates of size 250 per training iteration (instead of four updates of 500) and from skipping reward clipping. Hyperparameter values are detailed in Appendix A.3.

## 5.4 RESULTS

We show evaluation performance averaged across separate training runs (five for the less stable Atari games and three for the others). Each run uses a random seed never seen during development.

### 5.4.1 LB-DR VS BASELINE SAC/DDPG

Figure 2 compares lower bounding with discounted return (lb-DR) against SAC or DDPG baseline on Atari games and the episodic FetchEnv tasks.

For most tasks, lower bounding with episodic discounted return (lb-DR) performs similarly or better than the baselines. On Atari Breakout, Atlantis, Frostbite and Q*bert, and FetchPush and Fetch-PickAndPlace the gains are quite large. On Atari Seaquest, there is still a significant sample efficiency gain initially.

The lb-DR method is effective, but is it really due to improvements to the value targets? Figure 5 (Appendix A.4) looks at the fraction of training experience where lower bounded value target is actually higher than the baseline Bellman value target over the course of training. For the episodic FetchEnv tasks, as training progresses, a meaningful fraction of experience start to benefit from better value targets, and the average return performance also starts to improve over the baseline, although a large fraction of experience benefiting from higher value targets does not always mean a much higher average return (see FetchSlide). For most Atari games, improved value target does lead to significant performance gains, the only exception being Breakout, where value improvement does not immediately lead to performance gain.

### 5.4.2 LB-GD AND LB-DR+GD VS HER

Figure 3 compares lower bounding with goal distance return (lb-GD) and lower bounding with both goal distance and discounted return combined (lb-DR+GD) against the much stronger HER baseline, on the goal conditioned episodic FetchEnv and Pioneer tasks.

It seems on the easier FetchEnv tasks, lower bounding isn't able to outperform HER, but on the more challenging Pioneer Push and Reach tasks, lower bounding is able to achieve over 70% more sample efficiency. It seems the more complex the task, the wider the margin of gain.

We also looked at the fraction of experience where the lower bounding goal return is higher than the Bellman target (see Appendix A.4). It quickly grows to 1-8% and then slowly drops, matching the region where the new method outperforms the baselines in average return.

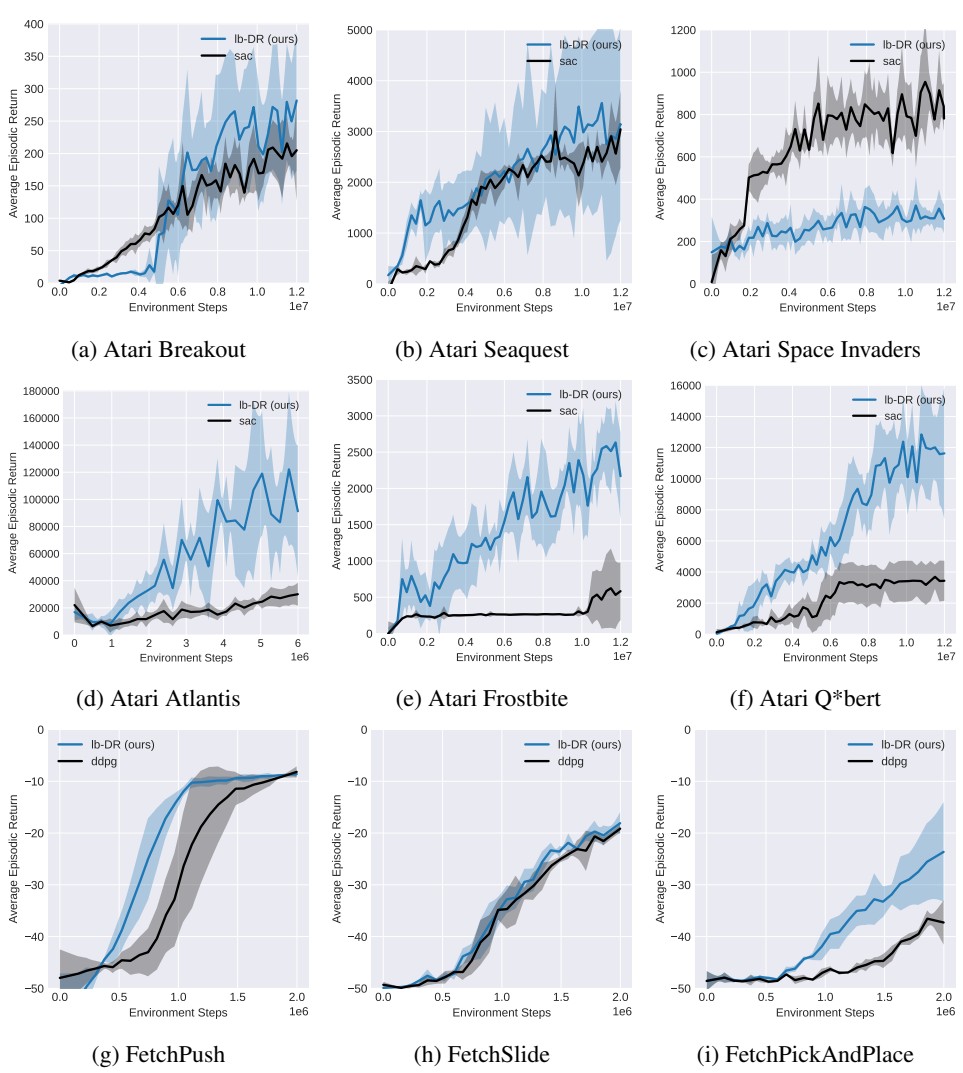

Figure 2: Evaluated average return of value target lower bounding with discounted return (lb-DR) vs SAC or DDPG on Atari games and episodic FetchEnv tasks. Solid curves are the mean across five (for Atari) or three (others) seeds, and shaded areas are +/- one standard deviation.

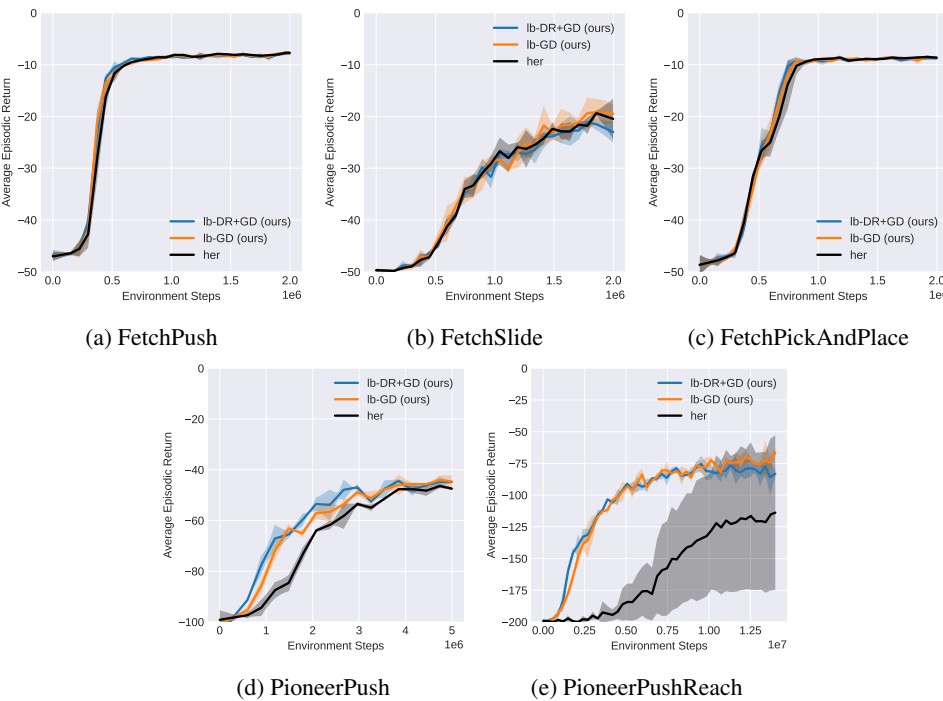

(a) FetchPush  (b) FetchSlide  (c) FetchPickAndPlace

(d) PioneerPush  (e) PioneerPushReach

Figure 3: Value target lower bounding with goal distance return (lb-GD) and lb-DR+GD vs HER on episodic FetchEnv and Pioneer tasks. Solid curves are the mean across three seeds, and shaded areas are +/- one standard deviation.

## 6 RELATED WORK

Prior works (Fujita et al., 2020; Hoppe & Toussaint, 2020; He et al., 2016; Oh et al., 2018; Tang, 2020) employed several different ways of computing future returns and using that as a lower bound to improve value learning. It is quite easy to introduce biases and inefficiencies into the process and end up with a suboptimal or inefficient algorithm. Our work is the first to point out that one efficient way of doing it, namely value target lower bounding, converges to the optimal value in the tabular case. We are the first to point out that the theory works generally, even for stochastic environments. We list several possible ways of computing the lower bound from training experience, which are true lower bounds only for deterministic environments, and demonstrate the effectiveness of such lower bounds in illustrative examples and experiments on a variety of tasks.

Fujita et al. (2020) used a method very similar to the lb-DR+GD variant, noted the limitation to deterministic tasks, and showed that value target lower bounding improved sample efficiency for a goal conditioned robotic grasping task. Hoppe & Toussaint (2020) similarly proposed to bound the value target using a simplified MDP with a subset of actions of the original MDP. Neither work gave any theoretical guarantee.

He et al. (2016) used empirical return with bootstrap to improve value learning. They integrate the lower (and upper) bounds as constraints when optimizing the Q function. Their method is more difficult to use due to an additional loss and hyperparameters to tune, and is more expensive to compute than directly lower bounding the value target. Their method needs to evaluate the value function on all future time steps. This severely limits how many time steps it can look ahead when computing discounted return. They evaluated on Atari games, showing higher sample efficiency than DQN, but appears worse than value target lower bounding on Breakout, probably due to looking ahead only four time steps. The limitation to deterministic tasks wasn't mentioned in the paper, (but is actually present due to the use of empirical return in computing the lower bound), and neither any convergence analysis. Appendix A.5 offers more discussions related to this method and n-step returns.

Our work is subtly but importantly different from the prior works on lower bound Q learning or Self Imitation Learning (SIL) (Oh et al., 2018; Tang, 2020). SIL uses empirical return $R$ to lower bound the value function itself (instead of the *value target*). This is achieved by adding an off policy value loss during on-policy (AC or PPO) training ($L_{value}^{sil} = \frac{1}{2}|v(s) - \max(v(s), R)|^2$). When the value function overestimates, the SIL value loss becomes zero, and keeps overestimating. Mixing the SIL loss with the loss from the baseline algorithms probably helped to correct the overestimation, but no theoretical guarantee was given. In evaluation, SIL was often compared to on-policy Actor Critic or PPO baselines, so it was not clear how much of the gain was due to lower bounding and how much due to off-policy value learning. In this work, we bound the Bellman value target (Equation 5), so overestimates are automatically corrected via Bellman updates, and convergence is guaranteed in the tabular case. We also use off-policy algorithms as baselines for a cleaner comparison.

Kumar et al. (2020) (DisCor) also recognized that bootstrapped value targets can be inaccurate. It impacts learning adversely under function approximation, while we handle the general case. DisCor uses distribution correction to sample experience with accurate bootstrap targets more frequently.

Interestingly, it is common practice to lower and upper bound the returns to the possible region, e.g. Andrychowicz et al. (2017) bounds value between $[-\frac{1}{1-\gamma}, 0]$.

## 7 CONCLUSIONS

In theory, value target lower bounding converges to the same optimal solution as the original Bellman value iteration. In practice, several ways of finding value lower bounds using empirical discounted return for deterministic episodic tasks are examined.

Precomputing discounted future return and storing into the replay buffer allows efficient lower bound computation, and can achieve much higher sample efficiency than baselines such as SAC, DDPG or TD3 in most tasks. The Appendix A.5 also includes comparisons against related methods such as td-lambda and Retrace.

Simple goal distance based return, requiring little extra space or compute, achieves large gains in certain long horizon tasks over HER, and performs similarly as HER in the simpler tasks.

### 7.1 FUTURE WORK

There are probably better ways of finding value lower bounds that speed up training even more. There may be ways of using bootstrapped value in computing the lower bound, for n-step return targets or for non-episodic tasks.

Estimating value lower bound for environments with stochastic transitions or rewards may be possible, e.g. by learning a reward function to help average out the randomness in the empirical return. Extending to partially observable environments would be harder but probably still doable.

Other ways of bounding the value target, e.g. upper bounding, may be worth investigating as well, e.g. to reduce overestimation in regions of poor reward.

### REPRODUCIBILITY STATEMENT

Our code change is based on a publicly available RL library, with strong baselines already implemented. Our relatively small code change is committed to a private github repository, which we plan to open source upon publication. When running experiments, the snapshot of the code used to run each experiment is stored together with the results. Experiment parameters are gin-configured and controlled by our automation script, with each experiment label corresponding to the set of configurations used for that experiment, so there is little room for manual error when running many experiments across different tasks, methods and hyperparameters.

Experiments are done in simulation with pseudo randomness. We've run our code on different machines with different GPU hardware using the same docker image, and the results are reproducible up to every float number using the same random seed. In a few cases, we've also run our code on different hardware and software (CUDA and pytorch), and the results are similar though not the same at the float number level.

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

# A  APPENDIX

## A.1  PROOF OF THEOREM 1

We want to prove that under the new operator $\mathcal{M}_f \circ \mathcal{B}$, the value function converges to the same optimal value function given by the Bellman operator $\mathcal{B}$.

*Proof.* Let $v^*$ be the fixed point and optimal value of the original Bellman operator: $v^* := \mathcal{B}^\infty(v)$, $v_1$ be any value function, and $s$ any state,

$$
\begin{aligned}
&|\mathcal{M}_f \circ \mathcal{B}(v_1)(s) - v^*(s)| \\
=&|\max(\mathcal{B}(v_1)(s), f(s)) - v^*(s)| \\
&\forall s \text{ where } f(s) > \mathcal{B}(v_1)(s): \\
&\text{above} = |f(s) - v^*(s)| = v^*(s) - f(s) < v^*(s) - \mathcal{B}(v_1)(s) = |\mathcal{B}(v_1)(s) - v^*(s)| \\
&\therefore \text{above} < |\mathcal{B}(v_1)(s) - v^*(s)| \\
&(\text{This is because } f \text{ lower bounds } v^* : v^*(s) \geq f(s) > B(v_1)(s).) \\
&\forall s \text{ where } f(s) \leq \mathcal{B}(v_1)(s): \\
&\text{above} = |\mathcal{B}(v_1)(s) - v^*(s)| \\
\leq&|\mathcal{B}(v_1)(s) - v^*(s)| \\
=&|\mathcal{B}(v_1)(s) - \mathcal{B}(v^*)(s)| \\
\leq&\gamma||v_1 - v^*||_\infty
\end{aligned}
$$

The last line above is because the Bellman operator $\mathcal{B}$ contracts at rate $\gamma$.

Hence, $||\mathcal{M}_f \circ \mathcal{B}(v_1) - v^*||_\infty = \max_s |\mathcal{M}_f \circ \mathcal{B}(v_1)(s) - v^*(s)| \leq \gamma||v_1 - v^*||_\infty$.

According to the definition of convergence to $v^*$, we need to find an $N$, such that $\forall \epsilon > 0$, $\forall v_1 \neq v^*$, $\forall n > N, ||(\mathcal{M}_f \circ \mathcal{B})^n(v_1) - v^*||_\infty < \epsilon$.

We can easily calculate that $N = \log_\gamma \frac{\epsilon}{||v_1 - v^*||_\infty}$ (note, $\gamma < 1$) satisfies the condition, which concludes the proof that any value function $v_1$ will converge to $v^*$ under the lower bounded Bellman operator $\mathcal{M}_f \circ \mathcal{B}$.

$\square$

This proof works for action values as well, by simply replacing the value function above $v(s)$ with the action value $q(s, a)$, and the value lower bound $f(s)$ with the action value lower bound $f(s, a)$.

## A.2  AN ILLUSTRATIVE EXAMPLE

Figure 4 includes a fairly general example showing how value target lower bounding would improve value learning. Suppose we enhance an off policy algorithm such as DDPG with value target lower bounding (lb-DR), when there is no training experience hitting the target state, no meaningful training happens for the baseline or lb-DR. However, when there is one trajectory hitting the target state, all states along the trajectory will soon be propagated with meaningful return, and nearby states will also enjoy faster learning. As the state space becomes larger and the time horizon longer, a successful trajectory will likely speed up learning quite a bit.

## A.3  HYPERPARAMETERS

Hyperparameters of the baseline algorithms follow published work in the case of FetchEnv (Plappert et al., 2018). For Atari and Pioneer Push and Reach tasks, they are tuned using one set of random seeds and after keeping the hyperparameters fixed, trained with a different set of random seeds and evaluated. We avoided tuning of the parameters of the baseline method for value target lower bounding, except for the Atari games where value target lower bounding learned a bit faster with slightly more frequent training updates (8 updates of 250 transitions per training iteration) than the baseline (4 updates of 500 transitions) and without reward clipping. For Atari Atlantis, Frostbite and Q*bert, we report results with reward clipping as it did not affect performance much.

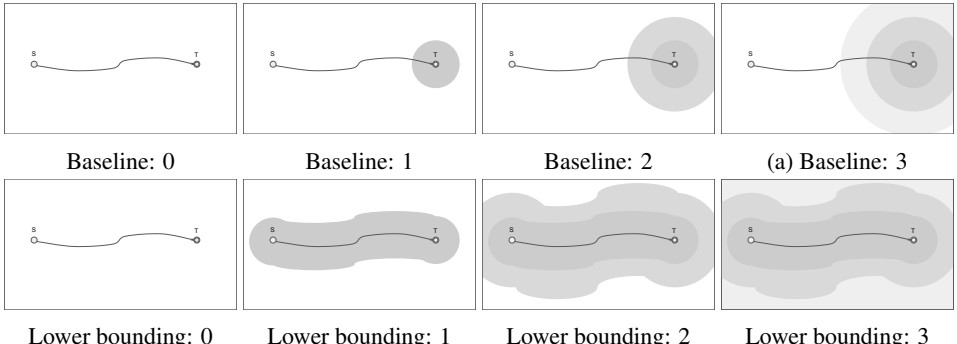

Figure 4: Illustration of value target lower bounding speeding up value learning as training progresses from stages 0 to 3. The task is to navigate in the state space from start state S to end state T, with sparse reward 1 at T and 0 elsewhere. The curve from S to T denotes a training experience that reaches the target. The shaded areas denote roughly states whose value has been significantly improved during training up to that stage.

Baseline parameters reported below are tuned using development random seeds and fixed during evaluation with a separate set of random seeds.

For the Atari games, DQN with only one training environment takes too long to train so we instead use SAC as baseline. There are no strong reported results, so we tuned the hyperparameters a bit and found it to outperform published Actor-Critic results on Atari Breakout (Oh et al., 2018). We use 30 environments, unrolling 8 time steps every training iteration, with each iteration containing 4 updates each with a minibatch of 500 transitions sampled from the 1 million time step replay buffer. 500 time steps are collected before training starts. Target networks are updated every 20 training updates. Discount $\gamma = 0.99$. The SAC target entropy is set to the entropy of uniformly distributing 0.1 probability mass across all but one actions. Actions are repeated deterministically for 4 frames (even for Space Invaders, despite 3 being used by Mnih et al. (2013)), and the latest 4 frames are stacked and rescaled to [-1, 1] to form the 84x84x4 input tensor. Rewards are clipped between -1 and 1. Network structures are the same as Double DQN (van Hasselt et al., 2015) with 3 convolution layers, with input layer 32 filters of size 8 stride 4, then 64 filters of size 4 stride 2 and 64 filters with size 3 stride 1, 1 fully connected of size 512 before output. We train for 12 million steps (48 million frames) for each task (except for Atlantis where episodes are very long and we only train for 6 million steps) and evaluate every 1000 iterations averaging across 100 episodes using $\epsilon$-greedy policy with 5% random actions.

For FetchEnv tasks, DDPG and HER learn faster than their TD3 variants and are reported here. Hyperparameters are the same as used by Plappert et al. (2018), with 38 parallel environments unrolling 50 time steps per train iteration, training 40 updates per iteration, targets are updated once every 40 updates. For each update, a minibatch of 5000 transitions are sampled from the replay buffer of size 2 million. Discount $\gamma = 0.98$. Actions are $\epsilon$-greedy with 30% random actions. 80% hindsight experience. Observations are normalized to have zero mean and unit variance based on the statistics of the recent observations.[3] Networks are 3 fully connected layers of size 256. Length of the episodes are capped at 50. We train for 2 million frames and evaluate every 40 iterations averaging across 200 episodes.

For Pioneer Push and Reach tasks, TD3 is used, (we simply equip DDPG with two critics for clipped double Q learning(Fujimoto et al., 2018)), which works better than DDPG with one critic. Parameters are mostly the same as in FetchEnv, except for using 30 parallel environments, 100 steps of unroll per training iteration, 6 million time step replay buffer, 50% hindsight experience, discount $\gamma = 0.99$ and not using observation normalization. Length of the episodes are capped at 100 for Push and 200 for Push and Reach. We train for 5 million frames for Push and 14 million for PushReach and evaluate every 200 iterations averaging across 100 episodes.

---

[3]We found that the use of observation normalization is critical in reproducing HER results on FetchEnv.

For FetchEnv and Pioneer tasks, the target networks are updated every 40 train updates softly, with weight 0.95 on the existing target network parameters and 0.05 on the incoming.

We use Adam optimizer with learning rate $5e^{-4}$ for the Atari games and $1e^{-3}$ for all others, and $\hat{\epsilon} = 1e^{-7}$ for all tasks.

## A.4 PLOTS

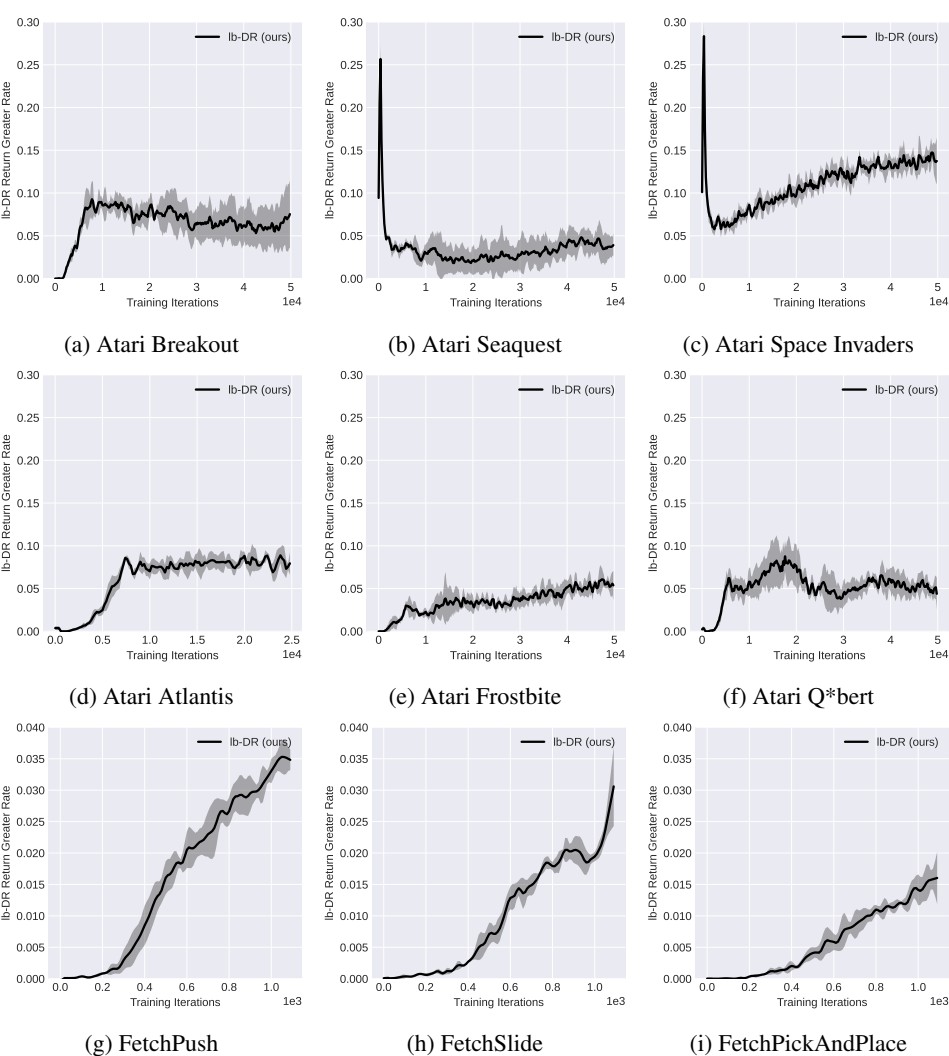

Figure 5: Fraction of training experience where lb-DR value target is greater than the Bellman target, on Atari games and episodic FetchEnv tasks, plotted against the number of training iterations. Solid curves are the mean across five (for Atari) or three (others) seeds, and shaded areas are +/- one standard deviation.

Figure 5 shows the fraction of training experience where lb-DR value target is greater than the Bellman target from SAC/DDPG. They correlate well with actual performance (Figure 2) and with how value is learning (Figure 6). For Atari Breakout the converged value is much higher than that of the baseline. It is unlike an overestimation, and is actually close to the average discounted return that we also summarized in tensorboard (omitted here).[4] The baseline value of 2 is actually very far away from its average discounted return of 25, even though its policy is already getting a reward of

---

[4]Back of envelope calculation: an episode of reward 400 evenly spread across its 1000 time steps with 0.99 step discount would lead to an average value of roughly $\frac{400/1000}{1-0.99} = 40$.

200 per episode. This is likely due to the inaccurate bootstrap values of the baseline method, and will probably take much longer to converge.

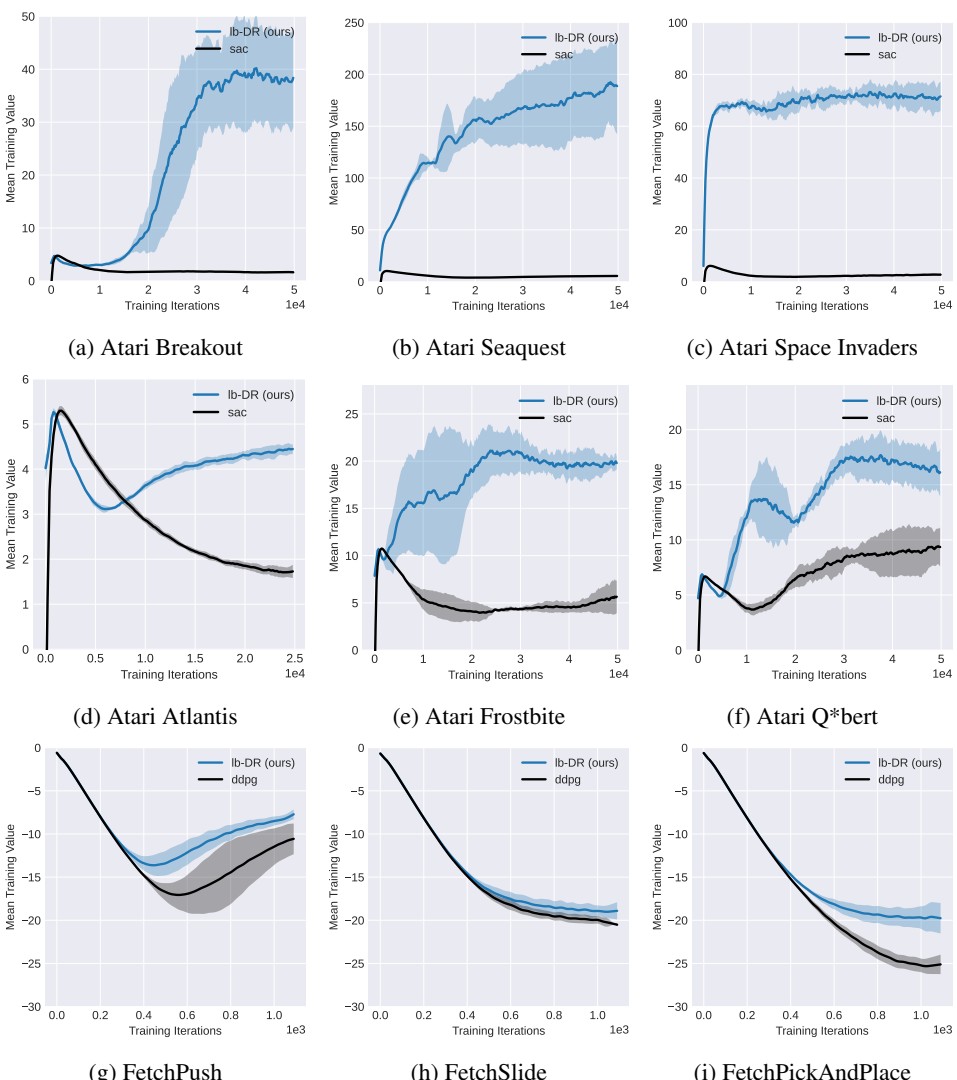

Figure 6: Learned values of lb-DR and SAC (for Atari games) and DDPG (for FetchEnv tasks), evaluated on the training experience and plotted against the number of training iterations. Solid curves are the mean across five (for Atari) or three (others) seeds, and shaded areas are +/- one standard deviation.

Figure 7 shows the fraction of training experience where the lb-GD is higher than the Bellman value target from HER, in the goal conditioned (episodic FetchEnv and Pioneer) tasks. It seems, for FetchEnv tasks, where lb-GD only performs slightly better than HER, the fraction of experience with improved value target is quite small (less than 1%). Hindsight relabeling is probably already producing fairly high value targets. For Pioneer Push and Reach tasks, lb-GD performs much better in average return, and the fraction of experience with higher value target is also much larger (peaking around 2-8%).

This again correlates well with the value learned, shown in Figure 8.

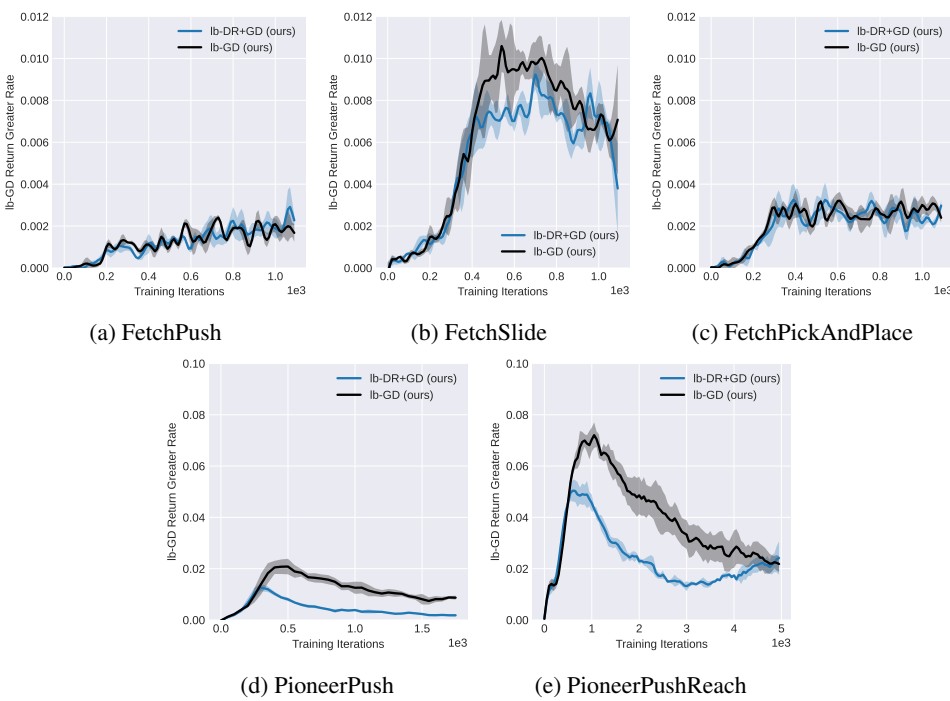

(a) FetchPush      (b) FetchSlide      (c) FetchPickAndPlace

(d) PioneerPush      (e) PioneerPushReach

Figure 7: Fraction of training experience where lb-GD or lb-DR+GD value target is greater than the Bellman target, on episodic FetchEnv and Pioneer tasks, plotted against the number of training iterations. Solid curves are the mean across three seeds, and shaded areas are +/- one standard deviation.

## A.5 N-STEP RETURN BASED METHODS

### A.5.1 N-STEP RETURN METHODS

We also experimented with n-step return, td-lambda return and Retrace (Munos et al., 2016) but decided to give up on the direction due to the following reasons:

1) We compared DDPG one-step return against DDPG with n-step return, td-lambda and Retrace on FetchPush, and found that a small n works similarly as the baseline one-step DDPG, and a larger n hurts training. This is likely due to the off-policy bias in n-step return causing the n-step estimate to be potentially worse than the one-step estimate, for example, when off-policy low return experiences are used to compute value targets. Introducing importance sampling weights (Retrace) would help reduce the bias, but at the same time significantly downweight the off-policy high return experiences, making an ineffective use of such experiences. The overall benefit of n-step methods is limited.

None of these issues are present in value target lower bounding: (a) It does not incur any off-policy bias, and (b) as long as an experience renders high reward, being off-policy does not affect its ability to improve the value target.

2) Computing n-step td-lambda return requires more computation due to evaluating value networks on all n-steps of the experience, and slows down training time significantly with a large n.

On the other hand, value target lower bounding precomputes and stores discounted return in the replay buffer, and incurs very little additional computation.

3) Tuning n-step return involves many hyperparameters like the number of steps n, the td-lambda parameter, replay buffer size and prioritized replay to expire old experiences and sample recent ones more frequently, target network update parameters to reduce potential overestimation, and parameters for importance sampling. But still, after all the tuning, it only slightly outperform one-step DDPG on FetchPush or SAC on Breakout, and is below the lower bounding method. For td-

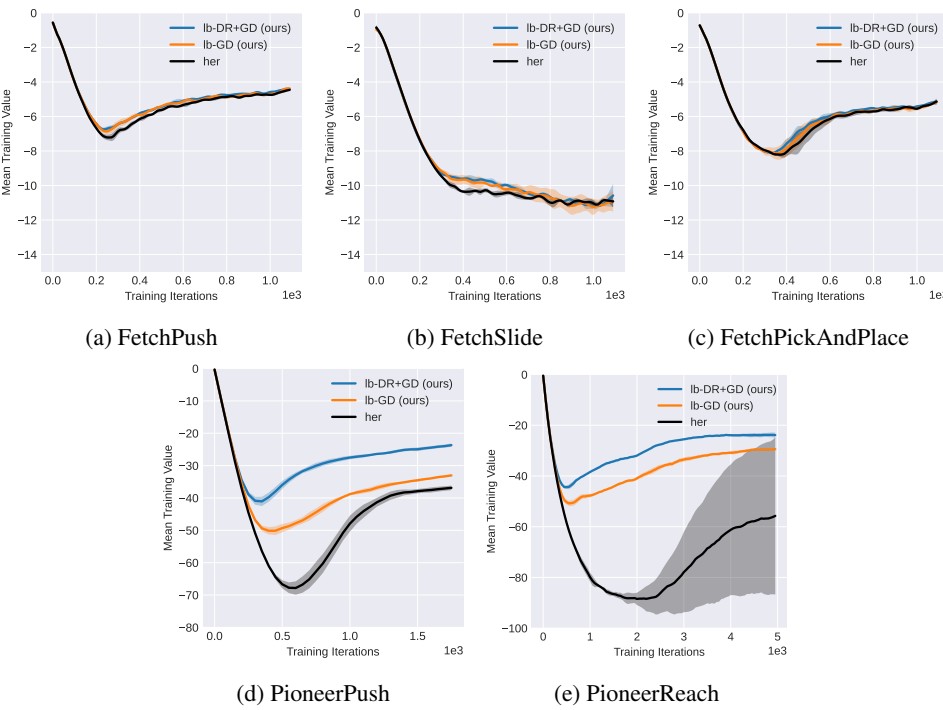

Figure 8: Learned values of lb-DR, lb-DR+GD and HER on episodic FetchEnv and Pioneer tasks, evaluated on the training experience and plotted against the number of training iterations. Solid curves are the mean across three seeds, and shaded areas are +/- one standard deviation.

lambda and Retrace, the best performance comes from 3-step td with $\lambda = 0.95$, replay buffer length 400k and all other parameters the same as the baseline DDPG or SAC. Retrace underperforming the baseline in Breakout is similarly observed in the original paper (Munos et al., 2016).

On the other hand, value target lower bounding requires no hyperparameter tuning, learns faster on most tasks and converges higher on some of the more difficult tasks.

4) Value target lower bounding can be applied on top of n-step return methods as well, so is more of an orthogonal problem.

Overall, n-step methods are much more expensive and difficult to use, and the much simpler and effective lower bounding method still maintains an advantage in effectiveness and performance. We show the performance comparison in Figure 9 with learned values in Figure 10.

### A.5.2 OPTIMALITY TIGHTENING WITH N-STEP RETURNS

He et al. (2016) use bootstrapped n-step return to lower and upper bound the value during training. They frame the problem as a constrained optimization problem where the distance between the value and the Bellman value target is minimized subject to the constraints that the value function must be within the lower (and upper) bounds. Their work is more general than the value target low bounding methods due to 1) including a value upper bound as well as lower bound, and 2) using bootstrapping, so it's applicable to non-episodic tasks as well.

Compared to value target lower bounding, several limitations exist.

1) The prior work bounds the value function itself (similar to lower bound q learning (Oh et al., 2018; Tang, 2020)), instead of bounding the Bellman value target. This could cause suboptimal training because the Bellman target itself could be outside the bounds, causing contradictory training targets and losses. Imagine the current value for a state is 1, its Bellman value target may be a low 0, and the lower bound may be a high 2, then it's unclear which way the value function should go.

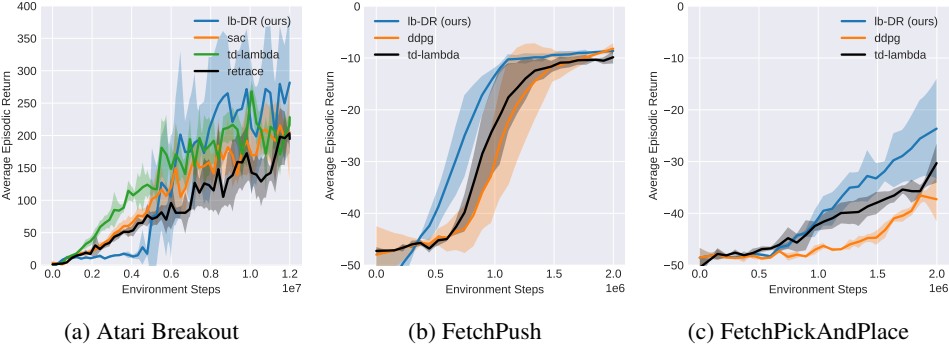

(a) Atari Breakout      (b) FetchPush      (c) FetchPickAndPlace

Figure 9: Evaluated average return of value target lower bounding with discounted return (lb-DR) vs SAC or DDPG, td-lambda and Retrace on Atari Breakout and episodic FetchEnv tasks. Solid curves are the mean across five (for Atari) or three (others) seeds, and shaded areas are +/- one standard deviation.

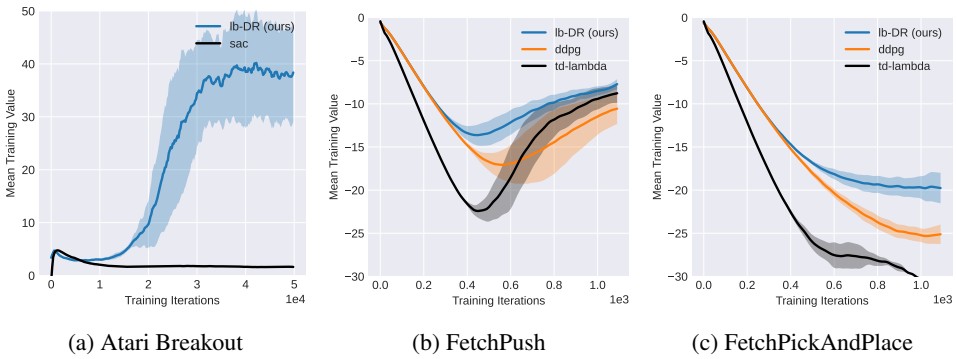

(a) Atari Breakout      (b) FetchPush      (c) FetchPickAndPlace

Figure 10: Learned values of lb-DR and SAC (for Atari games), DDPG (for FetchEnv tasks), td-lambda and Retrace, evaluated on the training experience and plotted against the number of training iterations. Solid curves are the mean across five (for Atari) or three (others) seeds, and shaded areas are +/- one standard deviation.

It will depend largely on the mixing weight between the two losses $\lambda$ and whether initial values overestimate, which can be hard to tune in practice.

2) The prior work does not include any theoretical analysis and misses the limitation to only deterministic tasks. The lower and upper bounds are in fact not correct bounds, even on deterministic tasks, because of the use of bootstrapped values together with the empirical discounted return.

3) In order to compute the bootstrapped values, the value network needs to be evaluated on all n future time steps, severely increasing GPU memory consumption and compute. Because of this increase in compute, in experiments, it could only look at a limited (4) timesteps into the future, while lb-DR can look all the way to the end of an episode with very little extra computation and storage.

We implemented the method (He et al., 2016) and integrated into our baselines. We ran on FetchPush and Atari Breakout, with hyperparameters number of time steps $n = 4$ and the penalty coefficient $\lambda = 4$, following the original paper.

We found that the prior method overestimates value a lot due to two reasons: a) taking max over the n-step returns for n from 1 to 4, and b) the use of the bootstrap value, causing the lower bound to be above what's actually achievable.

We also improved their method by lower bounding the Bellman value target with n-step return (with bootstrap) instead of imposing the constraint on the value function itself. But it still overestimates value and does not learn as quickly as the baseline one-step DDPG or SAC.

We also adjusted $\lambda$ to much lower values, hoping to control overestimation and improve over the baseline. Even with a very small lambda of 1e-7, it is still slower than DDPG baseline on FetchPush, likely because initial values are overestimates. On Atari Breakout, with a small lambda of 1e-7, it learns slightly faster than the SAC baseline but still way below the value target lower bounding method.

### A.6  A STOCHASTIC EXAMPLE

Using empirical return directly as value lower bound can lead to value overestimation, as shown in the stochastic MDP example below.

Assume state $S_0$ always goes to $S_1$, and $S_1$ gets reward $\pm 2$ 50% of the times. Then $v(S_0) = v(S_1) = 0$. However, with lower bounding, for the lucky case with reward 2, the value target for $S_0$ is $\gamma \max(2, v(S_1)) = 2\gamma$, and for the unlucky case with reward -2, the value target for $S_0$ is $\gamma \max(-2, v(S_1)) = \gamma v(S_1) = 0$. On average, $v(S_0)$ will be overestimated to be $\gamma$.

It is worth noting that lower bounding the action value directly as done in SIL (Oh et al., 2018) will overestimate $v(S_1)$ as well, whereas lower bounding the value target will produce the correct $v(S_1)$. This is because the same trajectory is used to both produce the Bellman value target ($\pm 2$ for $S_1$) and the lower bound ($\pm 2$ for $S_1$) which will be exactly the same for a given trajectory.

### A.7  DOES LOWER BOUNDING WITH EMPIRICAL RETURN REQUIRE THE POLICY TO BE DETERMINISTIC?

The use of empirical return to lower bound the optimal value does not require the policy to be deterministic. It does require the policy class to include the optimal policy (deterministic when the task is deterministic) or some policy that's close to the optimal policy. Otherwise, empirical return could still overestimate the optimal value achievable by the policy class. For example, Q learning assumes that the policy class includes the optimal policy which is the greedy one (and is deterministic). Because of that, the behavior policy can be non-deterministic and suboptimal, and it doesn't affect the learned value to reach optimality (as long as the behavior covers enough of the state space).

### A.8  POTENTIAL IMPROVEMENT

Note that the goal distance based return (lb-GD) of Section 3.1.1 is a very simple way of arriving at a reasonable lower bound with near zero additional computation. The bound could be made tighter. Typically, an $L_2$ distance threshold is used to judge goal achievement, which will likely be satisfied a few time steps before exactly arriving at the hindsight goal. To compute such a tighter bound would require evaluating the reward function across the trajectories of experience using all possible hindsight goal states, and storing them in the replay buffer, i.e. episode length squared more computation and more storage space. It may be worth doing when episodes are short, or doing it only for a small number of time steps into the future when e.g. rewards are non-negative.

