# OpenReview forum: "Faster Reinforcement Learning with Value Target Lower Bounding"
_ICLR.cc/2022/Conference — ICLR 2022 Submitted_

### Official Review · Reviewer_hHjM · 2021-10-20

**Correctness:** 2
**Technical Novelty And Significance:** 3
**Empirical Novelty And Significance:** 3
**Recommendation:** 6
**Confidence:** 5

**Main Review:**

Replacing my previous review which clearly was a misunderstanding on my end.

----

The work proposes a low bound on the value by considering the current sampled return. In deterministic domains this provides a lower bound on the optimal value function v^*. Under the correct assumptions this scheme can be shown to converge.

I think the work is interesting and the results are nice. Experiments themselves can be improved but are sufficient.

Main limitation is deterministic MDPs -- the world isn't deterministic and almost no task is. Randomness may be small but it is not non-existent.

**Summary Of The Paper:**

This work proposes to improve learning rate by lower-bounding the critic using observed trajectories.
Theoretically, in deterministic domains, such a scheme should improve learning rate as it grounds the value towards realistic regions.

**Summary Of The Review:**

Interesting work which can be of interest to the general RL community and lead to follow up research.

---

> ### Author Response · Authors · 2021-11-23
> **Replies to Reviewer hHjM**
>
> Thank you for the review, helpful comments and suggestions.
>
> The reviewer may have missed one key distinction: lower bounding the value function itself - $max(R, v(s))$ - which could overestimate value, versus lower bounding the Bellman value /target/ - $max(R, r(s, a) + \gamma * v(s’))$ - which is our approach and the theoretically proven one.  The math is in Section 4, and the distinction is discussed in the Related Works section about (Oh et al 2018 and Tang 2020).  We’ve since made this distinction more clear in Section 4, and added one example for the stochastic case in the Appendix which demonstrates its effect.  More replies are listed below.
>
>
> * Theoretical justification
>
> The reviewer seems to think that the theory requires the task to be deterministic, which is not correct.  The theoretical result in Section 2 does not rely on the task being deterministic, it works for the stochastic case as well.  Section 3 lists ways of using empirical return as lower bounds, which introduces the requirement for the task to be deterministic.
>
> We’ve made it more clear in Section 2 and 3.
>
>
> * Reliance on the policy being deterministic?
>
> The use of empirical return to lower bound the optimal value does not require the policy to be deterministic.  It does require the policy class to include the optimal policy (deterministic when the task is deterministic) or some policy that’s close to the optimal policy.  Otherwise, empirical return could still overestimate the optimal value achievable by the policy class.  For example, Q learning assumes that the policy class includes the optimal policy which is the greedy one (and is deterministic).  Because of that, the behavior policy can be non-deterministic and suboptimal, and it doesn’t affect the learned value to reach optimality (as long as the behavior covers enough of the state space).
>
> We added this discussion to the paper.
>
>
> * Empirical performance
>
> We fixed a bug in our code that caused overestimation of empirical discounted return for our method (the rewards used to compute the Bellman value targets were clipped to the range [-1, 1], but rewards used for computing episodic discounted return were not similarly clipped for the Atari games).  We also included more Atari games in the experiments.  Now the results on Atari look more prominent.
>
> For the other experiment setups, most setups show 70% or more sample efficiency gains over the baseline given a deterministic environment (DDPG on FetchEnv, HER on Pioneer tasks).  The only unclear setup is with HER as baseline on FetchEnv tasks, which is a fairly strong baseline for a relatively simple set of tasks.  (Sections 5.1.2 and 5.1.2 have details on why FetchEnv is simpler than the Pioneer setup.  It’s mostly due to a smaller range to move in, and a higher level relative position + PID control vs lower level force control.)
>
> Considering that value target lower bounding requires no hyperparameter tuning and very little additional computation, the gains are perhaps quite meaningful already.

---

> > ### Comment · Reviewer_hHjM · 2021-11-24
> > **Thank you for your response**
> >
> > Before I update my review, I have a final question.
> >
> > I understand that given some lower bound, the method will converge faster. However, saying that the sampled trajectory is a lower bound on the 1-step Bellman update is not so trivial.
> >
> > Do you have a proof for this? I was unable to find one in the paper.
> >
> > E.g., given a sampled trajectory {(s_0, a_0, r_0), ...., (s_N)} ~ pi
> > Then max(sum \gamma^t r_t, r_0 + \gamma * v(s_1)) will converge to the true value function for a fixed policy.
> > Or in other words given Theorem 1 that sum \gamma^t r_t lower bounds v(s).

---

> > > ### Author Response · Authors · 2021-11-24
> > > **Thank you for the question**
> > >
> > > The n-step discounted return (including bootstrap value) was used by He et al (2016), and is not exactly a lower bound due to bootstrapping.  In this work, we only used the cumulative discounted reward all the way to episode end as the lower bound.  We did not provide a proof, but it seems quite obvious that under deterministic environments, it is a strict lower bound of the optimal value. (Section 3.1 has details, and Related work also listed this difference.)
> > >
> > > Agree, it could be made more clear with a formula.  Fixed in our private draft.

---

> > > > ### Comment · Reviewer_hHjM · 2021-11-24
> > > > **True**
> > > >
> > > > Thank you, I agree on deterministic environments this is a lower bound. I'll update my review.

---

### Official Review · Reviewer_MXsM · 2021-11-01

**Correctness:** 3
**Technical Novelty And Significance:** 1
**Empirical Novelty And Significance:** 1
**Recommendation:** 3
**Confidence:** 5

**Main Review:**

This approach is not novel compared to say [1,2] and does not provide any meaningful insight, theoretical or empirical, over these existing papers.
- Theoretical results are obvious. It would be interesting if the authors could provide some theoretical results suggesting that this approach might be meaningfully better, such as faster convergence. Or discuss the case where rewards/environment is stochastic.
- Empirical benefits seem limited. There is no analysis or ablation over possible hyperparameters, different bounds, etc.

Minor:
- Notation for Theorem 1 is unexplained.
- Algorithm 1 is unnecessary.

References:
- [1] He, Frank S., et al. "Learning to Play in a Day: Faster Deep Reinforcement Learning by Optimality Tightening." (2016).
- [2] Tang, Yunhao. "Self-imitation learning via generalized lower bound q-learning." (2020).

**Summary Of The Paper:**

The authors use the online episodic return as a lower bound to the optimal value function and show that using this bound improves performance of deep RL baselines.

**Summary Of The Review:**

This approach is not novel and does not provide any meaningful insight, theoretical or empirical, over existing papers.

---

> ### Author Response · Authors · 2021-11-23
> **Replies to Reviewer MXsM**
>
> Thank you for your review, comments and suggestions.
>
> The reviewer may have missed one key distinction: lower bounding the value function itself - $max(R, v(s))$ - which could overestimate value, versus lower bounding the Bellman value /target/ - $max(R, r(s, a) + \gamma * v(s’))$ - which is our approach and the theoretically proven one.  The math is in Section 4, and the distinction is discussed in the Related Works section about (Oh et al 2018 and Tang 2020).  We’ve since made this distinction more clear in Section 4, and added one example for the stochastic case in the Appendix which demonstrates its effect.  More replies are listed below.
>
>
> [Replies to main concerns]
>
> * Related works
>
> [1] (He et al 2016)  Thank you for pointing out this citation.  Their work is definitely relevant and very interesting (includes a value upper bound as well as lower bound).  However, our work still brings a lot to the table.  The prior work doesn’t include any theoretical analysis and misses the limitation to deterministic tasks.  The lower and upper bounds are in fact not correct bounds, even on deterministic tasks, because of the use of bootstrapped values together with the empirical discounted return.  Their method requires tuning additional parameters and incurs more computation (due to evaluating the values of states on all n steps, similar to the td-lambda method).  We added a discussion of this in the paper.
>
> [2] (Tang 2020) is already cited and discussed together with the original lower bound q learning / SIL paper (Oh et al 2018).  Their mistake is to lower bound the value function instead of the Bellman value target, which could overestimate the value and converge to the wrong value.
>
> Overall, these two prior works highlight how easy it is to make a mistake here, and perhaps underscore the importance of the theory and the right way of doing value /target/ lower bounding, which is a contribution of this work.  The correct approach is also computationally more efficient.  We’ve made this more clear in the Related Work section.
>
>
> * Discussions of the stochastic case
>
> We’ve added discussions and examples on the stochastic case in the Appendix.
>
>
> * Faster convergence
>
> Section 2.2 (second point after the theorem) mentions the condition where faster convergence could be achieved, i.e. when lower bounding actually helps improve the value to get closer to the optimal value in L-inf norm.  The theorem guarantees convergence to the optimal value, at a rate at least as fast as the baseline.  Simply improving the value function isn’t enough for faster convergence.  It has to happen in a region that matters to the L-inf distance between the current value and the optimal value, i.e. where the gap between the two are largest.
>
> We’ve made the text more clear and also added a fairly general illustrative example showcasing when it could learn faster in the Appendix A.2.
>
>
> * Empirical benefits
>
> We fixed an overestimation bug for Atari (the rewards used to compute the Bellman value targets were clipped to the range [-1, 1], but rewards used for computing episodic discounted return were not similarly clipped for the Atari games) and included more Atari games.  Results look more prominent now.  On most deterministic task setups, SAC on Atari Breakout, DDPG on FetchEnv tasks, HER on Pioneer tasks, there are clear (close to 100%) sample efficiency gains, and with higher converged value on some of the harder tasks.  The only setup with unclear gains is HER on FetchEnv, which is a strong baseline on a relatively simple set of tasks.  A more detailed discussion of this is in Sections 5.1.2 and 5.1.3.
>
> Considering that value target lower bounding requires no hyperparameter tuning and very little additional computation, the gains are perhaps quite meaningful already.
>
>
> * Lacking of hyperparameter ablation
>
> There may be some misunderstanding here.  The proposed lb-DR and lb-GD methods do not have any hyperparameter to tune.  However, it is a valid point to add more investigation to show how and why this method works.  Since the first submission and given requests from other reviewers, we’ve added illustrative examples, plots of learned value to show when the method works and to more confidently confirm that it is due to value target improvement.  We’ve also made the hyperparameter section more clear to avoid any potential confusion.
>
>
> [Replies to maybe minor points]
>
> * Removing Algorithm 1
>
> We are hesitant to remove it, as we find it clarifies the discussion and makes it easier on some of the readers.
>
>
> * Theorem 1 notation
>
> The Bellman operator is defined in Equation (1), the lower bounded Bellman operator $\mathcal{M}_f \circ \mathcal{B} (v)$ is defined inside the theorem.  $B^{\infty}(v)$ is defined in the theorem to denote the optimal value, and is basically applying the operator an infinite number of times.  We are happy to clarify any other points that may seem unclear.

---

> > ### Comment · Reviewer_MXsM · 2021-11-30
> > **Reply to Authors**
> >
> > Thank you for the response, and I apologize for not being more involved during the end of the discussion period. I appreciate the number of changes by the authors. I have looked through the adjustments made by the author to the paper as well as the other reviews. I stand by my original review.
> >
> > My suggestion for improving the paper for either camera-ready or future submission would be to mainly focus on more extensive empirical evaluation.
> > - While including additional baselines is helpful, the empirical results over the base algorithm are limited. Given the simplicity of the technique, I think its essential to examine it on multiple algorithms and more environments. The performance benefits of the technique seem very minimal in most cases. If this improvement is consistent across many algorithms and environment, then it is meaningful. I don't think I can infer that from the current set of experiments.
> > - In general, the choice of baseline is unusual. For example, the authors add the lower bound to SAC for the Atari tasks, even though SAC is a continuous control method (there are discrete extensions but this is non-standard), and then don't use SAC for the continuous control tasks. I think it would be important to use SOTA baselines such as DQN or its variants as a discrete action baseline, and SAC for continuous control to show this technique "matters".
> > - Adding more "content". There is almost no substance to the paper (one theorem and some empirical results presented in large figures). This can be additional theoretical results, more extensive empirical evaluations, analysis, etc. I'm not convinced this paper adds anything meaningful to the literature in the current state. While the authors have pointed out differences from their algorithm to He et al, it mostly comes down to minor details, so I’m not sure we can say this is a novel technique and even if it outperforms He et al on a small selections of environments, He et al is not a fundamental or widely-used method in the literature.
> > - This hasn't affected my score, but in general the language/writing is very informal/casual, and the paper is missing a sense of polish.

---

### Official Review · Reviewer_e2TJ · 2021-11-02

**Correctness:** 4
**Technical Novelty And Significance:** 3
**Empirical Novelty And Significance:** 3
**Recommendation:** 6
**Confidence:** 4

**Main Review:**

**Significance**: The question of how to combing Monte Carlo return estimates with value functions is quite important to the design of good RL algorithms. While prior work has proposed a number of methods for doing this (e.g., TD($\lambda$), Retrace, n-step returns), the proposed method has been relatively under-explored (though, see [1]). However, this paper does not compare to these alternative methods, so it is unclear if the proposed method is better than these prior methods. Moreoever, while the proposed method only is guaranteed to work well in deterministic environments, prior work has shown that these alternative approaches do work well in stochastic environments. I'd highly recommend that the paper be revised to include a comparison against TD($\lambda$), Retrace and n-step returns.

**Originality**: The proposed method is similar to [1]. I would recommend that the paper be revised to discuss and compare to this prior work.

**Clarity**: The paper is generally clear. I'd recommend reorganizing the sections as follows: Introduction, Related Work, Background, Lower Bound, Algorithm, Results.

**Correctness**: All claims seem correct. While the paper notes that the proposed method isn't guaranteed to perform well in stochastic environments, I would recommend emphasizing this claim. For example, here's a toy example that could be included to illustrate this issue:
> Assume a bandit problem with 2 actions, one of which gets reward 1 and the other gets reward $\pm2$. While the correct action is the first action, I believe that the proposed method will end up selecting the second (incorrect) action.
* Is it possible to provide a guarantee that the proposed method won't perform that much worse than the optimal policy on stochastic environments?
* "deterministic versions of the three games" -- Is this the standard evaluation protocol? I was under the impression that prior work usually added some randomness.
* The hyperparameter $\alpha$ is set to $0.0005 \approx 0$ for the Atari results. Since $\alpha = 0$ corresponds to the baseline, I am a bit concerned that the difference between the baseline is just random fluctuations (perhaps caused by trying multiple hyperparameters for the proposed method without correcting for the multiple testing problem).

**Minor comments**:

* "Temporal Difference" -> "temporal difference"
* The introduction lacks structure. I'd recommend making sure that each paragraph is intended to explain a single idea.
* "seems to converge ... "seems straightforward" -- The use of "seems" makes these statements appear informal.
* Sec. 2.2 -- I like that the general idea of value target lower bounding is introduced before the complete method.
* "at least as fast ... because it's more likely..." -- Clarify whether these are theoretical or empirical claims.
* "which we encourage you, the reader, to come up with" -- Avoid the use of the second person ("you") in formal technical writing. Perhaps: "future work may investigate alternative lower bounds."
* "The FetchEnv tasks" -- Add a citation.
* Most of the details in Sec. 5.3.1 can be moved to the appendix. I'd recommend shortening this section to ~5 lines.
* The future work discussed in the conclusion is great!
* A.1 -- The math at the end of the proof is so terse that it is very hard to parse.


[1] He, Frank S., et al. "Learning to play in a day: Faster deep reinforcement learning by optimality tightening." arXiv preprint arXiv:1611.01606 (2016).

**Summary Of The Paper:**

This paper proposes a new RL algorithm based on a modified Bellman backup equation. The main idea is to estimate the value of a state in multiple ways (using a Q function and using Monte Carlo returns) and then to take the maximum over these estimates. The paper shows that, if all estimators are lower bounds on the true value, then the proposed method converges to the optimal policy. Experiments confirm that the proposed method outperforms standard Q learning (i.e., with regular Bellman backups) on some tasks.

**Summary Of The Review:**

This paper is studying an important problem, but is missing a discussion and comparison to a number of prior works. I therefore am voting to reject the paper.

-------------------------
Updates during/after review period:
* 11/23: Increasing score 3 -> 5 after new experiments showed that the proposed method outperforms other value estimation techniques.
* 11/24: Increasing score 5 -> 6 after clarification of difference with (He 2016).

---

> ### Author Response · Authors · 2021-11-23
> **Replies to Reviewer e2TJ**
>
> Thank you for the careful review, the helpful citation, examples, comments and suggestions.
>
> [Replies to main concerns]
>
> * Related work [1] (He et al 2016)
>
> Thank you for the citation.  It is interesting - includes a value upper bound as well as lower bound, and uses bootstrapping, so it’s applicable to non-episodic tasks as well.  However, the value target lower bounding method still brings a lot to the table:
>
> 1) The prior work bounds the value function itself (similar to lower bound q learning (Oh et al 2018; Tang 2020)), instead of bounding the Bellman value /target/.  This could cause suboptimal training because the Bellman target itself could be outside the bounds, causing contradictory training targets and losses.
>
> 2) The prior work does not include any theoretical analysis and misses the limitation to only deterministic tasks.  The lower and upper bounds are in fact not correct bounds, even on deterministic tasks, because of the use of bootstrapped values together with the empirical discounted return.
>
> 3) Because of the more compute and GPU memory needed to evaluate the value of n future time steps, it only looks at a limited (4) timesteps into the future, while lb-DR can look all the way to the end of an episode.
>
>
> We implemented [1] and ran on FetchPush and Atari Breakout and found that the prior method overestimates value a lot and destroys performance.  We added a detailed discussion in the Appendix.
>
>
> * Comparisons with n-step TD methods
>
> The reviewer pointed out that the benefit of the method could be largely due to propagating return multiple steps from the future, and should compare against n-step return methods.
>
> This is a very good point.  The authors had the same concern, but gave up on the direction early on, due to some of the reasons listed below.  After receiving this review, we did more investigation, extensive tuning and more comparisons, and here are the results:
>
>   1) N-step methods are typically limited to a small n, due to estimation biases and computational constraints.
>       On the other hand, value target lower bounding precomputes and stores discounted return in the replay buffer, and incurs very little additional computation.
>   2) Tuning n-step return involves many hyperparameters.  But still, after all the tuning, it only slightly outperform one-step DDPG on FetchPush or SAC on Breakout, and is below the lower bounding method.
>       On the other hand, value target lower bounding requires no hyperparameter tuning, learns faster on most tasks and converges higher on some of the more difficult tasks.
>   3) Value target lower bounding can be applied on top of n-step return methods as well, so is more of an orthogonal problem.
>
> We included experimental results and detailed discussion in the Appendix.
>
>
> * Stochastic environments
>
> ** Bandit example
>
> Thanks for bringing up this simple example.  It needs to be modified to work, because for the lucky experience of taking the second action with reward 2, there will always be the unlucky experience of taking the second action with reward -2.  During lower bounding, the lucky experience will use 2 as value target, and the unlucky experience will use the return -2 lower bounded with empirical future return -2 as value target, which yields the correct target -2.  Overall, the value of the second action should end up being 0 correctly.  For this example, lower bound q learning or SIL (Oh et al., 2018; Tang, 2020) will fail, because SIL lower bounds the Q value (leading to q(a2) as value target for a2), while we lower bound the value target.
>
> We’ve added the modified example to the Appendix of the paper.
>
> ** Deterministic versions of Atari games
>
> Yes, when reporting results, people by default report on the deterministic version (GameName-v4).  For example, the prior works cited https://github.com/junhyukoh/self-imitation-learning uses GameName-v4.  DDQN (Hasselt et al 2015) uses the deterministic versions but with random restart from human playing.  We used a random number of (at most 30) noop steps after each game reset, before handing it to the agent, which causes the game’s initial conditions to be different, and is standard practice.
>
>
> * Hyperparameter \alpha
>
> We removed the use of \alpha after fixing the bug that overestimated future discounted return (the rewards used to compute the Bellman value targets were clipped to the range [-1, 1], but rewards used for computing episodic discounted return were not similarly clipped for the Atari games).
>
> * Moving Related Work section earlier
>
> We’ve considered doing it, but find it hard to get into the technical details which are only introduced later in the paper.  Without these details (concepts such as value vs value target, lower bounding, implementation of the lb-DR variant etc.), proper discussions of the related works would be hard to achieve.
>
>
> [Replies to maybe minor concerns]
>
> Thank you for the detailed suggestions.  Fixed all applicable.

---

> > ### Comment · Reviewer_e2TJ · 2021-11-23
> > **Clarifications**
> >
> > Thanks for the rebuttal. Is the following summary of the revisions correct?
> >
> > 1. New experiments show that the proposed method outperforms He 2016.
> > 2. New experiments show that the proposed method outperforms n-step returns.
> > 3. New experiments show that the proposed method outperforms n-step returns.
> > 4. A bug fix means that the proposed method now works with $\alpha = 1$ on all tasks.

---

> > > ### Author Response · Authors · 2021-11-23
> > > **Yes**
> > >
> > > 1. Yes, we implemented He 2016 on SAC, and were not able to make it outperform the baseline SAC, despite unit tests and tuning related hyperparameters.  We did not include plots, but explained what we did, and an explanation of why we think that happened in Appendix.
> > > 2. Yes.  Plots in Appendix.
> > > 3. Yes, assuming you mean Retrace.  Plots in Appendix.
> > > 4. Yes.

---

> > > > ### Comment · Reviewer_e2TJ · 2021-11-23
> > > > **Score updated: 3 -> 5**
> > > >
> > > > Thanks for the clarifications. I've updated my score to 5.
> > > >
> > > > Could the authors explain a bit more the differences with (He, 2016)? Showing an equation that compares the two TD targets might be helpful.

---

> > > > > ### Author Response · Authors · 2021-11-24
> > > > > **differences with (He et al 2016)**
> > > > >
> > > > > Thank you for your questions.
> > > > >
> > > > > There are mainly two differences: 1) the way the lower bound is integrated into value training, and 2) how the lower bound is estimated.
> > > > >
> > > > > For 1), He et al (2016) use the lower bound (and upper bound) as constraints of the value function when doing value learning: minimizing $||Q(s, a) - \hat{Q}(s, a)||$ subject to $Q(s, a) \ge Q_{lower-bound}$, where $\hat{Q}$ is the Bellman value target.  In value target lower bounding, what's being lower bounded is the Bellman value target, not the value function, so the loss is $||Q(s, a) - \max(Q_{lower-bound}, \hat{Q}(s, a))$ (see Section 4).
> > > > >
> > > > > For 2), He et al (2016) used empirical return of n steps plus the bootstrap value of the last state $Q_{lower-bound} = \sum_{t=0..n-1}{\gamma^t r(s_t, a_t)} + \gamma^n * \bar{V}(s_n)$, while we used the empirical return all the way to the end of the episode, without using the bootstrap value for state $s_n$.
> > > > >
> > > > > Appendix A.5.2 has more details on the comparison.

---

> > > > > > ### Comment · Reviewer_e2TJ · 2021-11-24
> > > > > > **Thanks for the clarification**
> > > > > >
> > > > > > Thanks for the clarification. This now makes sense to me.

---

### Official Review · Reviewer_cupU · 2021-11-02

**Correctness:** 4
**Technical Novelty And Significance:** 3
**Empirical Novelty And Significance:** 3
**Recommendation:** 6
**Confidence:** 4

**Main Review:**

### Strengths

1. The paper presents a simple and interesting idea. Specifically, using a value function lower bound to modify the Bellman backup is intuitive and seems simple enough that future work could build on the idea effectively. Moreover, this idea is (to my knowledge) novel.
2. The proof of the main theorem seems correct. This result gives at least a useful sanity check that the proposed algorithm is reasonable.
3. The paper is in general clearly written. It does a good job of conveying a simple idea simply.

### Weaknesses

1. The paper provides no analysis proving or even clearly demonstrating faster convergence. The main motivation for the proposed algorithm is faster convergence. So, it would strengthen the paper greatly to prove this, even under some simplifying assumptions. For example, what conditions on the lower bound would yield provably faster convergence? It would even strengthen the paper to just offer some tabular examples that could be computed in closed form where the lower bound clearly improves convergence speed, but a more general proof would be even better. Also, perhaps some larger scale examples could be created where there still are clear and obvious benefits to the proposed approach.
2. The proposed lower bounds cannot handle stochasticity in the environment. This limitation is clear in the paper, but it a substantial limitation. For the idea to be more broadly applicable, there would need to be a way to handle stochastic environments.
3. This is a more minor point, but the algorithm as proposed in the paper only works with actor-critic algorithms that use SARSA-style value estimation. This is not made particularly clear in the paper, but is an important point since it causes the authors to use SAC instead of a DQN-based algorithm on Atari, which is non-standard. However, I don't immediately see why there is not a way to make the algorithm work with max-Q backups as in DQN. Perhaps the authors can comment on whether the algorithm can be made to work with a DQN-style base algorithm?
4. There are a few weaknesses with the experiments:
   1. The improvement of the proposed method over the baseline is just not very clear in most of the experiments. Often there is no noticeable difference except in the fetch pick-place task and pioneer push-react task. In Breakout the proposed method actually seems to learn slower initially, which seems to run counter to the motivation for the method.
   2. The value of $ \alpha $ seems to be a bit strange and needs some explanation. First, unless I missed something, the paper only lists the value of $ \alpha $ for the Atari games where it was 0.0005. What value was used in the other tasks? Second, if the value for $ \alpha $ is so small, it is not clear that the proposed algorithm is making a substantial difference at all. Why is such a small value used? Is the algorithm very unstable for larger values?
   3. In Figure 3 the values on the y-axis should be multiplied by 100 if the results are reporting percentages. Otherwise they should be labeled as the "fraction" of experience instead of percentage.
5. This is a more minor thought, but I am somewhat worried that the proposed approach could cause some instability in learning and this might explain why the experimental results are not so strong. Specifically, it is well established that overestimation of the value function is often a problem (and motivated Double DQN as well as TD3). By introducing the lower bound, this problem is potentially exacerbated since the new algorithm can only increase the values being backed up. So while this may seem to learn faster, it could also increase the instability. This also may be related to why such small values of $ \alpha$ are needed. Can the authors comment on whether they observe overestimation when using their method?

**Summary Of The Paper:**

This paper proposed value target lower bounding as a simple modification to the Bellman operator that intends to improve convergence speed. It proves that using such a lower bound in the Bellman backup does not change the fixed point in the tabular setting. The paper then proposes two instantiations of particular value target lower bounds: first by using the return in deterministic episodic MDPs and second by hindsight relabeling in goal-directed tasks. Finally, the paper offers some experiments using the proposed lower bounds.

**Summary Of The Review:**

Overall, I like the idea of the value function lower bound, but don't think the paper is ready for publication yet. The paper could benefit from some tighter analysis and better examples showing the benefit of the algorithm. Moreover, the current experimental results are not quite convincing as explained above.

---

> ### Author Response · Authors · 2021-11-23
> **Replies to Reviewer cupU**
>
> Thank you for the careful review, and the helpful comments and suggestions.
>
> [Replies to main concerns]
>
> * Limitation on deterministic tasks
>
> The reviewer recommends extending the techniques to stochastic environments before publication.
>
> The authors share the same concern, but still pushed for publication for the following reasons:
>
>   1) The value target lower bounding theory and proof works generally, even for stochastic environments.  It is the direct use of empirical return as a lower bound that is limited to the deterministic case.  (Reviewer seems to be aware of this already, but it’s worth stressing.)
>
>   2) Many widely used benchmarks are deterministic.  The Atari games have deterministic versions that are widely used.  Many simulated robotic control tasks e.g. the FetchEnv tasks are deterministic or near deterministic.  So there is a material benefit in bringing the technique to the attention of RL practitioners sooner rather than later.
>
>   3) Some of the advanced RL methods, e.g. MuZero (Nature, Vol 588, 24/31 December 2020, Page 606) and Self Imitation Learning (Oh et al., 2018; Tang, 2020), are also limited to deterministic tasks, and can fail on simple stochastic cases.  We do not intend to compare this work directly with MuZero here, but just want to make sure the reviewer is aware when making the assessment of publishing vs delaying.
>
> Overall, we think the benefit of spreading the knowledge of this simple and effective method probably outweighs the limitation to deterministic environments.
>
>
> * Faster convergence
>
> Section 2.2 (second point after the theorem) mentions the condition where faster convergence could be achieved, i.e. when lower bounding actually helps improve the value to get closer to the optimal value in L-inf norm.  The theorem guarantees convergence to the optimal value, at a rate at least as fast as the baseline.  Simply improving the value function isn’t enough for faster convergence.  It has to happen in a region that matters to the L-inf distance between the current value and the optimal value, i.e. where the gap between the two are largest.
>
> We’ve made the text more clear and added an illustrative example to demonstrate when faster learning could happen.
>
>
> * Illustrative example for faster learning
>
> We’ve added an illustrative example (Appendix A.2) to show that value target lower bounding could speed up learning in a relatively general setting.  We hope this addresses most of the concern on why and when the proposed method works.
>
>
> [Replies to maybe minor points]
>
> * DQN style value targets
>
> DQN style value targets are still bootstrapped, and can be similarly improved with lower bounding.  We’ve added it into the introduction.  The reason we did not use DQN as a baseline is due to its very long wall clock time, having only one environment for rollout.  With SAC, we have 30 environments rolling out in parallel during training.
>
>
> * Value overestimation
>
> The reviewer thinks that the new algorithm can only increase the values being backed up and could potentially lead to value overestimation and learning instability.
>
> Theoretically, as the lower bounds are strictly below optimal value, overestimation shouldn’t be a problem.  In practice, we looked at the learned value, and did not observe any obvious overestimation for the deterministic environments.  They are inline with the actual discounted return averaged across states.  We’ve included more discussion and plots of learned values for all the tasks in the Appendix of the new revision.
>
>
> * Experiments: empirical performance over the baseline not very clear
>
> We fixed a bug in our code that caused overestimation of empirical discounted return for our method (the rewards used to compute the Bellman value targets were clipped to the range [-1, 1], but rewards used for computing episodic discounted return were not similarly clipped for the Atari games).  We also included more Atari games in the experiments.  Now the results on Atari look more prominent.
>
> For the other experiment setups, most setups show 70% or more sample efficiency gains over the baseline given a deterministic environment (DDPG on FetchEnv, HER on Pioneer tasks).  The only unclear setup is with HER as baseline on FetchEnv tasks, which is a fairly strong baseline for a relatively simple set of tasks.  (Sections 5.1.2 and 5.1.2 have details on why FetchEnv is simpler than the Pioneer setup.  It’s mostly due to a smaller range to move in, and a higher level relative position + PID control vs lower level force control.)
>
>
> * Experiments: value of the mixing weight \alpha
>
> We've completely removed \alpha after fixing the return estimation bug.
>
>
> * Experiments: Figure 3: “fraction” vs “percentage” of experience
>
> Good catch, fixed throughout the paper.

---

> > ### Comment · Reviewer_cupU · 2021-11-26
> > **Post-rebuttal comments**
> >
> > Thanks for following up on the issues raised in my initial review.
> >
> > The bug fix in the experiments that removes alpha and substantially improves the empirical results has convinced me to revise my score from a 3 to a 6. I always liked the basic idea and I think the improved execution makes the main argument of the paper more convincing.
> >
> > On the issue of stochastic environments, I agree that this is not strictly required and that the theory goes through for the stochastic case. Without a way to implement the algorithm in stochastic settings this is still a weakness of the proposed approach, but not a fatal one.
> >
> > I still think it would be nice if the convergence speed improvement could be quantified into a simple lemma, but the text after Theorem 1 and the example provided in the appendix make the point more clearly than in the first version of the paper. This example should perhaps also be pointed to in the text after Theorem 1 directly. Moving this example to the main text could also be valuable (and perhaps some of the text describing the experimental setup could be moved to the appendix).
> >
> > Some of the other reviewers raised some issues with novelty, but I think this has been adequately addressed.

---

> > > ### Author Response · Authors · 2021-11-27
> > > **Thanks**
> > >
> > > Thank you for the quick reply and suggestions.
> > >
> > > Updated our private draft:
> > > 1) moved the example/illustration to the main text, (moved Experiment Setup details to Appendix)
> > > 2) pointed to the example after Theorem 1.
> > >
> > > Glad the example helped improve understanding.

---

### Public Comment · ~Yasuhiro_Fujita1 · 2021-11-09
**On value target lower bounding**

For your information, if I understand the proposed trick correctly, we used the same trick in our paper [1], which has been accepted at IROS 2020 (see Paragraph e of Section IV.A). To be clear, we applied it to QT-Opt with HER for efficient learning in sparse-reward multi-goal settings. Our work differ from the submission in that we applied it to QT-Opt, not DDPG, TD3, or SAC, and that we did simply lowerbound a value target with a discounted return (calculated in hindsight) without introducing a mixing weight.

[1] Y. Fujita, K. Uenishi, A. Ummadisingu, P. Nagarajan, S. Masuda, and M. Y. Castro, “Distributed Reinforcement Learning of Targeted Grasping with Active Vision for Mobile Manipulators,” IROS, 2020. https://arxiv.org/abs/2007.08082

---

> ### Author Response · Authors · 2021-11-23
> **Thank you**
>
> Thank you Yasuhiro for bringing our attention to this nicely written paper.  We’ve added the citation and discussions to the Method sections and the Related Works section.

---

### Decision · Program_Chairs · 2022-01-20

**Decision:**

Reject

**Comment:**

In the end, this paper essentially proposes a minor variation on an idea that 1) has been published before, 2) is not used extensively at all, and 3) seems applicable (in its current form) only on deterministic environments.  This, without additional insights or analyses, seems too marginal a contribution for acceptance.

The paper is not poorly executed, and the authors engaged well during discussion, for which I would like to thank them.  I would like to encourage the authors to consider the reviewers comments, and in particularly perhaps answer more clearly and directly what they are adding to the literature.  It could be that there is something particularly insightful in the detailed differences with past work, but this has not become sufficiently clear to me during this discussion phase.